# Molecular Epidemiology, Virulence Traits and Antimicrobial Resistance Signatures of *Aeromonas* spp. in the Critically Endangered *Iberochondrostoma lusitanicum* Follow Geographical and Seasonal Patterns

**DOI:** 10.3390/antibiotics10070759

**Published:** 2021-06-22

**Authors:** Miguel L. Grilo, Sara Isidoro, Lélia Chambel, Carolina S. Marques, Tiago A. Marques, Carla Sousa-Santos, Joana I. Robalo, Manuela Oliveira

**Affiliations:** 1CIISA—Centro de Investigação Interdisciplinar em Sanidade Animal, Faculdade de Medicina Veterinária, Universidade de Lisboa, 1300-477 Lisbon, Portugal; s.isidoro10@gmail.com; 2MARE—Marine and Environmental Sciences Centre, ISPA—Instituto Universitário de Ciências Psicológicas, Sociais e da Vida, 1149-041 Lisbon, Portugal; carla.santos@ispa.pt (C.S.-S.); jrobalo@ispa.pt (J.I.R.); 3BioISI—Biosystems and Integrative Sciences Institute, Faculdade de Ciências, Universidade de Lisboa, 1749-016 Lisbon, Portugal; lmchambel@fc.ul.pt; 4Departamento de Biologia Animal, Centro de Estatística e Aplicações, Universidade de Lisboa, 1749-016 Lisbon, Portugal; carolinasegmarques@gmail.com (C.S.M.); tiago.marques@st-andrews.ac.uk (T.A.M.); 5Centre for Research into Ecological & Environmental Modelling, University of St Andrews, St Andrews KY16 9LZ, UK

**Keywords:** *Aeromonas*, antimicrobial resistance, virulence, typing, leuciscid, fish

## Abstract

Despite the fact that freshwater fish populations are experiencing severe declines worldwide, our knowledge on the interaction between endangered populations and pathogenic agents remains scarce. In this study, we investigated the prevalence and structure of *Aeromonas* communities isolated from the critically endangered *Iberochondrostoma lusitanicum,* a model species for threatened Iberian leuciscids, as well as health parameters in this species. Additionally, we evaluated the virulence profiles, antimicrobial resistance signatures and genomic relationships of the *Aeromonas* isolates. Lesion prevalence, extension and body condition were deeply affected by location and seasonality, with poorer performances in the dry season. *Aeromonas* composition shifted among seasons and was also different across river streams. The pathogenic potential of the isolates significantly increased during the dry season. Additionally, isolates displaying clinically relevant antimicrobial resistance phenotypes (carbapenem and fluroquinolone resistance) were detected. As it inhabits intermittent rivers, often reduced to disconnected pools during the summer, the dry season is a critical period for *I. lusitanicum*, with lower general health status and a higher potential of infection by *Aeromonas* spp. Habitat quality seems a determining factor on the sustainable development of this fish species. Also, these individuals act as reservoirs of important antimicrobial resistant bacteria with potential implications for public health.

## 1. Introduction

Past geological events that shaped freshwater landscapes in the Iberian Peninsula and promoted its geographical isolation greatly contributed to the high level of endemism currently observed in native leuciscid fishes (Teleostei: Leuciscidae) [1]. In spite of this biotic richness, leuciscids’ populations are following the decreasing trends observed in freshwater species worldwide [2], and around 70% of the known species in the region belonging to this taxonomic group present some level of threat to their conservation [3]. Several attempts are currently implemented to counter current trends, including habitat restoration and ex-situ breeding programs targeting several species [4].

Different factors play a key role as potential threats for the sustainability of leuciscid species. Namely, habitat degradation and destruction due to anthropogenic actions such as the construction of dams and weirs, water abstraction (e.g., for agriculture), river architecture alteration, increased pollution levels due to direct or land run-offs, and the introduction and proliferation of exotic species are known to impact these species’ survival [5]. For species living in the Mediterranean area, climate fluctuations are another key issue and are suggested to become more important in the future. Leuciscids that occur in intermittent streams of the southern part of the Iberian Peninsula are typically subjected to cyclical shifts between winter floods and extremely low water levels during the summer. In the dry season, the incidence of droughts increases in these streams, fragmenting the river and restricting movement and habitat availability for fishes [6]. This situation is likely to get worse with predicted climatic changes [7].

The loss of river connectivity seasonally imposed by the lowering water levels in the region often results in a series of more or less disconnected summer pools. It is common to observe a high number of individuals gathered in these small habitats where, due to a higher depth, water remains available—although with no flow, higher temperatures and lower dissolved oxygen levels [8]. Higher fish densities likely contribute to the deterioration of water quality parameters and to an increase of intra and inter-species competition for resources and individual contact; which potentially influences host’s immune function, with obvious consequences for overall fitness [6,9,10]. Despite these conditions being ideal for the spread of several pathogenic agents, the prevalence and impact infectious diseases might have in these populations is not clear so far. Investigations regarding the prevalence of specific pathogens are limited and often do not focus on the impact these agents might have for the host. Also, past studies have focused on parasitological and fungal surveys [11,12], neglecting the influence that other pathogenic agents—i.e., bacteria and virus—might have in leuciscids.

Bacterial pathogens are common infectious agents in fish species and assume particular relevance in this context [13]. When the water temperature increases, which is associated with the higher fish densities observed in the summer pools, microbial loads also tend to increase [14]. Members of the genus *Aeromonas* can be important in this situation, since infections caused by these bacteria are considered one of the most common type of bacterial infection detected in freshwater fish species, causing morbidity and mortality in both farmed and wild fishes [15]. Indeed, they are commonly known pathogenic agents ubiquitous in several aquatic ecosystems and are also accounted as both zoonotic agents and antibiotic resistance and virulence reservoirs [16,17,18,19].

It is important to understand the host-pathogen dynamics of these agents and how they can impact these already imperiled populations and constitute significant threats to their conservation. Identifying specific pathogens that might contribute to the decrease of a fish species’ stock and understanding how these pathogen communities are shaped by environmental conditions is the first step in creating species and habitat recovery plans that take into account specific susceptibilities for a population in particular and help to implement strategies to restore habitats, fighting detrimental impacts pathogenic agents might have.

Developing active sampling schemes, using subsets of individuals, can help to investigate both the prevalence and impact of selected pathogens—e.g., *Aeromonas* spp.—and produce valuable information for the interested stakeholders. Moreover, patterns among different populations from the same species occupying diverse geographical areas or between different seasons can potentially be discriminated during these sampling actions and result in the definition of risk factors which would contribute to a better management of the species. Additionally, and since there is a growing concern about the role that aquatic ecosystems play as end-point collectors of determinants of antibiotic resistance and virulence originating from land [20], using such sampling schemes as opportunistic actions to survey and detect antimicrobial resistance and virulence in natural aquatic habitats is of major importance to signal and prevent dissemination points between the interface land-water [21]. The microbiota of wild species is of particular significance since wildlife plays an active role in the dissemination of microbiota and genetic determinants in natural environments [22]. Also, *Aeromonas* spp. are known to express a great variety of virulence factors involved in host colonization and invasion, constituting a possible virulence reservoir in the aquatic environment [23]. Additionally, the potential of *Aeromonas* spp. to acquire and disseminate resistance determinants is widely acknowledged, making them relevant indicators of antimicrobial resistance in aquatic environments [24].

We hypothesize that *Aeromonas* communities associated with distinct populations of endangered leuciscids will vary according to the geographical origin of the populations—i.e., different exposure to environmental and anthropogenic conditions—and season (dry and wet). In addition, we suggest that distinct patterns of fish fitness (e.g., body condition and skin conformation) are present in populations of the same species. Also, we hypothesize that antimicrobial resistance signatures and virulence traits of the *Aeromonas* isolates vary according to different factors (i.e., location and season). To test these ideas, we used isolated wild populations of the critically endangered boga portuguesa (*Iberochondrostoma lusitanicum*; Collares-Pereira, 1980), a model species for endangered leuciscids, evaluated their skin lesion scores, body condition and *Aeromonas* spp. prevalence, composition, similarity relationships, virulence profiles and antimicrobial resistance signatures of the isolates.

## 2. Results

The fishing efforts in this study allowed for sampling 406 *I. lusitanicum* individuals in both seasons and all four locations (dry season—285; wet season—121). Furthermore, formation of groups based on age stratification (adults and juveniles) was achieved in both sampling campaigns.

Lesion prevalence in sampled individuals varied according to season, being significantly lower in the wet season (*p* < 0.001). In the individuals collected during sampling actions, there was a significantly higher proportion of fishes displaying lesions (*p* < 0.001) when compared to those who did not display any lesions. Among sampled locations, when compared to the Laje location, significantly lower levels of lesion prevalence were observed in individuals originating from Lizandro (*p* = 0.002) and significantly higher levels of lesion prevalence in individuals originating from Samarra (*p* < 0.001). 

The sampled rivers presented differences regarding flow regime. Water physical and chemical parameters, as well as site characterization, are presented in Table 1.

Body condition scores (BCS) differed significantly regarding location (*p* < 0.001). Individuals originating from Laje presented higher BCS, followed by those from Lizandro, Jamor and Samarra. No association between BCS and sampling season was found (*p* = 0.104), although a trend was observed with higher scores in the wet season. There was a positive correlation between BCS and the size of the animal (r = 0.46; *p* < 0.001).

Skin lesion score (SLS) of the individuals was significantly different between seasons and locations (*p* < 0.001). SLS was significantly lower in the wet season. Individuals from Samarra river presented higher SLS values than individuals from other locations. There was a negative correlation between individual’s size and SLS (r = −0.28; *p* < 0.001) and a negative correlation between SLS and BCS (r = −0.17; *p* = 0.015).

*Aeromonas* spp. prevalence was high in individuals from both seasons and locations (dry season = 100%, wet season = 92.2%). In total, a bacterial library of 376 isolates of *Aeromonas* was established. After clone elimination, 223 isolates were further studied. Species composition differed among locations and seasons (*p* < 0.001), revealing specific structures for each river in each season (Figure 1). In general, *A. hydrophila* was the most prevalent species isolated from the sampled animals in both seasons, despite reaching higher prevalence values in those from the dry season, followed by *A. veronii* and *A. media*. In the individuals sampled in the dry season, a predominance of *A. hydrophila* was observed in the Jamor river, outnumbering other species, while fishes from Samarra river presented higher levels of *A. veronii*. In the wet season, despite high levels of *A. hydrophila* in the sampled animals, individuals sampled in Jamor presented higher levels of *A. caviae*. On the other hand, animals from Laje presented higher values of *A. media* and *A. veronii*. The prevalence of *A. veronii* in fishes from Samarra was lower than the prevalence registered in the dry season.

Virulence indexes were significantly influenced by sampling season and the species of the isolates (*p* < 0.001). Isolates collected in the animals from the wet season scored significantly lower than those collected in animals from the dry season. Regarding the species of *Aeromonas*, *A. caviae* isolates presented the higher virulence indexes, followed by *Aeromonas* spp., *A. hydrophila*, *A. veronii* and *A. media*. No correlation between the virulence index and the multiple antibiotic resistance (MAR) index was found (r = −0.12, *p* = 0.081).

All the tested isolates (*n* = 222) displayed lipolytic activity. A total of 204 isolates (92%) had hemolytic activity, 89% of the isolates produced DNase activity, 86% exhibited proteolytic activity and 40% of the isolates were slime producers. Only 24% of the isolates exhibited gelatinase activity. 

Regarding the virulence signatures registered for isolates from each sampling location, shifts occurred for selected virulence factors between isolates from different locations and across seasons (Figure 2). Slime production varied significantly between isolates from different locations (*p* < 0.001), with higher prevalence in those from samples collected in Samarra, while the majority of the slime-producing strains were *A. veronii*. DNase prevalence was lower in isolates collected in the wet season (*p* < 0.001) and influenced by the origin (*p* < 0.001), with lower positive prevalence in those sampled in Laje river. Most of the DNase-negative isolates obtained in samples from this river were *A. media*. Regarding gelatinase prevalence, a higher positive prevalence was recorded on isolates from the dry season and from Lizandro and Jamor (*p* < 0.001). The vast majority of these isolates were *A. hydrophila*. For hemolytic activity, only season affected the prevalence of positive results (*p* = 0.012), with isolates from samples collected in the dry season presenting more positive results. Regarding protease, the sampling season influenced the prevalence of positive results (*p* = 0.024), since the isolates from dry season samples presented a higher prevalence of this virulence factor.

Among the studied bacterial collection, 30.2% (*n* = 67) of the isolates were considered multidrug resistant and none showed resistance to all the antimicrobials tested. The prevalence of multidrug resistant was significantly influenced by sampling season (*p* < 0.001), with a higher prevalence of such isolates being obtained from samples collected in the wet season (73.1%, *n* = 49). Similarly, the MAR index of the isolates also differed significantly among seasons (*p* < 0.001), with isolates collected from samples during the wet season presenting significantly higher scores. Isolates collected from samples from Jamor river presented the higher values, followed by those collected in Laje, Samarra and Lizandro. The MAR index values for the different *Aeromonas* species did not differ significantly (*p* = 0.837).

Higher levels of non-susceptibility (intermediate and resistant categories) were recorded for erythromycin (100%), amoxicilin/clavulanic acid (98%) and streptomycin (61%). Moderate levels were registered for enrofloxacin (40%), sulfamethoxazole/trimethropim (34%), imipenem (34%), amikacin (22%) and ceftazidime (19%). The lowest non-susceptibility levels were recorded for tetracycline (11%), florfenicol (8%), nitrofurantoin (8%) and aztreonam (4%).

Regarding the isolates’ resistance profile for each antimicrobial tested, fluctuations in the relative proportion of each susceptibility category occurred for the evaluated panel of antibiotics, both between locations and seasons (Figure 3). Regarding the results for ceftazidime, there was a higher prevalence of isolates presenting a non-susceptible status in samples from the wet season (*p* < 0.001) when compared to the dry season and a lower prevalence of non-susceptible status in those from Lizandro (*p* = 0.027) when compared to Jamor. In relation to enrofloxacin, similar trends to ceftazidime regarding season (*p* < 0.001) and location (*p* = 0.023) were observed. Regarding florfenicol (*p* = 0.006), nitrofurantoin (*p* = 0.015), tetracycline (*p* = 0.002) and sulfamethoxazole/trimethropim (*p* < 0.001), a higher prevalence of isolates with a non-susceptible status were registered in the wet season. For streptomycin, a lower prevalence of isolates with non-susceptible status occurred in Laje (*p* = 0.011) when compared to Jamor.

Molecular typing of the isolates and their relations evaluated by dendrogram analysis revealed a high number of clusters at the 66.98% of similarity (based on a joint evaluation of fingerprinting patterns, *Aeromonas* species, river of origin of the isolate and season in which was collected) among the studied collection (Appendix A). Several clones were identified at the 87.6% similarity (reproducibility level), corresponding to isolates originating from the same animal or from different animals in the same river. However, no persistence of clones was observed for samples obtained in the same location across seasons nor the presence of clones originating from samples from different rivers.

Regarding each season, specific trends were observed (Figure 4). In the dry season, four clusters were formed (based on the fingerprinting patterns, *Aeromonas* species and river of origin of the isolates). Cluster I, defined at 14.36% similarity, has 47 isolates and is mainly composed by isolates of *A. veronii*, with the presence of *A. hydrophila* and *A. media* isolates. The majority of isolates in this cluster was obtained from individuals from Samarra river. Cluster II, defined at 23.12% similarity, has 68 isolates and is mainly composed by *A. hydrophila* isolates, although *A. caviae*, *A. media* and *Aeromonas* sp. were also present. Origin of the isolates was more disperse, although a big contribution from samples from animals collected in Laje river is noted. Cluster III, defined at 32.91% similarity, has 13 isolates and is exclusively formed by *A. hydrophila* from animals from Lizandro river, and is mainly composed (84.6%) by clones retrieved from four distinct *I. lusitanicum*’s individuals. Cluster IV, defined at 38.43%, has 32 isolates and is mainly composed of *A. hydrophila* isolates, with a small proportion of *A. caviae* and *A. veronii*. The vast majority of the isolates were collected from animals in Jamor river, where clones of *A. hydrophila* (11 isolates) were recovered from six individuals.

In the wet season, eight clusters were determined (following the same parameters used for dry season). Cluster I, defined at 25.26% similarity, has 20 isolates and is mainly composed of *A. veronii* isolates, with additional *A. hydrophila*, *A. media* and *Aeromonas* sp. isolates. The majority were obtained from animals collected in Laje river. Cluster II, defined at 41.18% similarity, has 39 isolates and is mainly constituted by *A. hydrophila* isolates, with few *A. caviae* isolates. The vast majority of the isolates originated from samples collected in Jamor river. Cluster III, defined at 42.8% similarity, has 44 isolates and groups mainly *A. hydrophila* isolates with *A. media*, *A. veronii* and *Aeromonas* sp. also included. The isolates were obtained from fishes of the four rivers almost proportionally. Cluster IV, defined at 44.08% similarity, has eight isolates, all identified as *A. hydrophila* from samples collected in Lizandro river. Cluster V, defined at 26.31% similarity, has six isolates and is mainly composed of *A. veronii* and one isolate of *A. media*. The majority of the isolates were from samples collected in Laje river. Cluster VI, defined at 24.3% similarity, has 40 isolates and a predominance of *A. media* and *A. hydrophila*, but also includes *A. veronii* and *Aeromonas* sp. Samples collected in Laje and Samarra rivers contributed to a higher proportion of the isolates. Cluster VII, defined at 44.54% similarity, has four isolates, including *A. veronii*, *A. hydrophila* and *A. media*, from samples collected in Lizandro, Jamor and Samarra rivers. Cluster VIII, defined at 50.28% similarity, has six isolates, half of them corresponding to *A. media*, but also including *Aeromonas* sp. and *A. hydrophila*. All of the isolates from animals sampled in the Laje river.

When investigating isolates’ genomic relationships in each location, it is possible to form clusters that reflect the sampling season (Figure 5). That is particularly obvious for the Jamor river, where most of the isolates grouped in two clusters, each with a majority of isolates for one same sampling season. While such separation was also present in the other sampling locations, some clusters presented isolates belonging to both seasons.

Regarding the genomic relationships of the identified *Aeromonas* species, some trends were observed (Appendix A). Isolates of *A. caviae*, *A. hydrophila* and *A. veronii* presented clusters composed of a majority of isolates collected during a specific sampling season. This differentiation was particularly obvious in the case of *A. caviae*. Such differentiation was not present in the dendrogram of *A. media* isolates, although a low prevalence of this species in the dry season was recorded. Such grouping was not visible regarding the origin of the isolates, although some clusters were in their vast majority represented by isolates from samples collected in the same river (e.g., *A. hydrophila* in Jamor and Lizandro, *A. veronii* in Samarra, Laje and Lizandro).

Based on the cluster evaluation (Appendix A), Simpson’s diversity index did not differ across seasons (*p* = 0.448). Diversity indexes were considered high in both seasons (DDry season = 0.97; DWet season = 0.96). The indexes differed between locations, namely the diversity indexes recorded in Laje were significantly higher than those in the Jamor river (*p* = 0.011; Appendix A).

Regarding the analysis of the association between bacterial clusters and fish characteristics, the presence of a non-conventional data structure hinders conclusive results (namely, the fact that several isolates were obtained from the same fish, which induces a dependence across isolates, and hence different clusters within fish). There is absence of evidence that any given cluster lead to higher or lower values of SLS or BCS.

## 3. Discussion

Surveillance programs have the potential to unravel biotic and abiotic drivers of pathogen presence, composition and host-pathogen interaction. Such schemes can produce valuable information often lacking in natural habitat management plans. Although skin lesions in fishes can have a multifactorial origin, with several possible different agents involved and contribution of *Aeromonas* spp. to the lesions observed in *I. lusitanicum* cannot be concluded, the methodology applied in this study can access dynamics of bacterial pathogens with relevance for conservation medicine and, at the same time, serve as an opportunistic basis for the study of antimicrobial resistance and virulence prevalence and temporal shifts with a particular relevance for public health. In this study, we show that distinct populations of the critically endangered leuciscid fish *I. lusitanicum* display differential levels of general health with shifts according to seasons and locations, and that those modifications are accompanied by changes in the *Aeromonas* structure associated with them. Furthermore, the structure, antimicrobial resistance profile and virulence signatures of these *Aeromonas* isolates are shaped by seasonal and spatial drivers. Finally, we consider that *I. lusitanicum* populations are acting as reservoirs of isolates with clinically important antibiotic resistance phenotypes.

### 3.1. Body Condition Score (BCS) and Skin Lesion Score (SLS)

A higher prevalence of lesions and higher scores of SLS were detected in the dry season, and most of the individuals sampled during this season displayed skin lesions. This high incidence of cases reinforces the idea that the dry season, when several threats to freshwater fish have a simultaneous and cumulative negative impact [5], constitutes a temporal frame of fragility for the fitness and consequent survival of this endangered species. Fish skin responds quickly to changes in the environment [25], and several stressors are known to trigger the cellular damage that results in skin lesions such as those observed in this study [26]. Namely, oxygen depletion, exposure to xenobiotic chemicals and biotoxins, pH and temperature fluctuations, high levels of organic matter, parasites, stress and confinement are known to be correlated with the onset of skin diseases, both in wild and cultured fishes [25,27,28,29,30,31,32,33]. Additionally, and once skin damage is established and the homeostatic mechanism of the host is affected, optimal grounds for bacterial colonization and invasion are established, favouring the proliferation of opportunistic pathogens such as *Aeromonas* spp. [28]. This phenomenon was already observed in combination with a variety of stressors [27,34,35] and, in the case of motile aeromonads, likely benefits from warmer temperatures that potentiate bacterial proliferation [36].

Like skin lesions, fishes’ body condition also varies according to the stressors present in their environment. Growth and body condition have been reported to be negatively correlated with reduced water quality, pollution levels in the environment, lower prey abundance and habitat degradation [31,32,37,38,39]. Interestingly, although *I. lusitanicum* suggested prey items, such as macroinvertebrates and zooplankton, are available in the dry season [40,41], their diversity and abundance decrease in streams highly impacted by pollution and hydric stress [42,43]. This situation likely generates intra-specific and inter-specific competition for food, particularly with invasive species that are better adapted to environmental critical conditions [44,45]. It is also likely that, as observed for other species, *I. lusitanicum* individuals would change their dietary focus into plant material and detritus under critical conditions [46]. This negative dietary balance experienced by fish individuals will affect their ability for disease resistance, since a balanced nutritional intake is key for the immune system competence [47].

Both SLS and BCS were correlated with individual size: while smaller individuals displayed in average the highest SLS, they were also the ones scoring the lowest regarding their BCS. Such findings highlight the vulnerability of younger life stages towards the adverse conditions (i.e., overcrowding, decrease levels of food availability, predation) they encounter in their natural habitat, coupled with an immature immune system that does not allow them to fully prevent disease by pathogenic agents [48]. For this reason, bacterial infections in early life stages of fish are common [49]. Also, SLS levels were negatively correlated with BCS. Although causality direction is difficult to establish in this case, this finding suggests that homeostasis disruption in *I. lusitanicum* individuals is accompanied by overall deleterious effects that can compromise fish survival.

A marked differentiation in SLS and BCS occurred between animals sampled in different locations. In the Samarra population, a consistent higher prevalence of lesions and SLS was detected in the sampled individuals, and the BCS was also the lowest in average from the animals in all sampled locations. Contrarily, in fishes from Lizandro, the prevalence of lesions was the lowest. River characteristics, both regarding water quality and anthropogenic influences, seem to shape the individual homeostatic state and characterize the onset of dysregulations that culminate in the physical expression of lesions. Although, in general, all the studied streams present medium to low water quality indexes [50,51,52], the Samarra River presented the higher levels of total dissolved solids and electrical conductivity, suggestive of higher content of organic matter. In fact, in its last kilometres, this river is influenced by effluents from a wastewater treatment plant [52]. Bad water quality, restricted habitats (reduced summer pools) and habitat degradation are possible synergetic drivers that influence the overall health of *I. lusitanicum* and likely result in the observed changes for the Samarra population. In the Lizandro River, despite being influenced by agricultural runoffs [53], it seems that the sampled location provides a habitat compatible with the sustainable development of a *I. lusitanicum* population. These findings highlight the role that habitat quality have in the prevalence of healthy *I. lusitanicum* populations.

### 3.2. Aeromonas Prevalence, Structure, Similarity Relationships and Diversity

Regarding *Aeromonas* prevalence, a high level of isolates was detected in animals sampled in both seasons and across all locations. *Aeromonas* spp. prevalence in the aquatic habitat seems to be highly dependent on the climacteric conditions associated with the water [54]. Several studies report a predominance of *Aeromonas* prevalence during winter, with a significant decrease of bacterial levels during the summertime [54,55]. This is likely the case in regions where a strong temperature fluctuation is noted between seasons, while in temperate regions the opposite occurs and the climacteric conditions favour the establishment of bacterial populations all-year round, with a predominance when water temperature rises during summer months [56,57,58,59,60]. However, it has been proposed that *Aeromonas* prevalence in fishes remains stable throughout the year [54], as observed in our study. This is particularly important since fishes, besides being affected by pathogenic aeromonads, can also constitute reservoirs and maintain *Aeromonas* communities during adverse climacteric conditions. Even in the case that a determined bacterial strain fails to induce disease, an individual can constitute an important disseminator for other conspecifics that might be experiencing decreased or immature immunological functions (e.g., juveniles) and establish outbreaks in a population. This possibility is supported by our findings that demonstrate the sharing of bacterial clones among several individuals in each location, which is likely a consequence of the gregarious behaviour of *I. lusitanicum*.

In our study, the predominant *Aeromonas* species detected in *I. lusitanicum* were *A. hydrophila*, *A. veronii* and *A. media*. *A. caviae* was detected in a lower proportion. These findings are in agreement with previous results of prevalence studies conducted with fishes [61,62,63,64]. However, some studies report differences in *Aeromonas* prevalence in fish species [65,66,67,68,69]. This is probably a reflection of differential environmental pools of *Aeromonas* communities to which these fish species are exposed, coupled with host genotype adaptations that lead to particular associations of host and microbiota. The role of *A. veronii* as a primary pathogen involved in motile aeromonad septicaemia has been emphasized, which has been attributed to the role aerolysin has in the virulence of this species [67]. In our study, a predominance of *A. veronii* was recorded for individuals belonging to the Samarra river’s population during the dry season. The highest prevalence of skin lesions and the highest scores of SLS were also recorded in this population during the dry season, establishing a possible link with *A. veronii* prevalence. It is noteworthy that *A. veronii* may be linked with primary cases of infection in these individuals, contradicting the idea that *A. hydrophila* is the most virulent species for aquatic animals. *A. caviae* has previously been appointed as dominant in waters contaminated with sewage and is considered an indicator of faecal contamination [66,70]. Interestingly, in our study this species was only isolated in individuals from Jamor and Laje streams, the two locations associated with urban settings. It seems that the recovery of *A. caviae* from these locations exposes these streams as more impacted by sewage disposal, a likely consequence of urbanization [70].

Our results differ from other studies focusing on the prevalence and diversity of *Aeromonas* species in freshwater streams [60,66,70,71,72,73]. However, and although the aquatic environmental compartment is the most likely route for the acquisition of these isolates, the *Aeromonas* communities colonizing *I. lusitanicum* individuals probably differs in proportion regarding their environmental counter partners. Chaix et al. [60] showed that the *Aeromonas* communities associated with copepods differed regarding the communities available in the water column. This phenomenon mirrors bacterial adaptations regarding host colonization, signaling the success determined strains have in adhering to and colonizing animal hosts when compared with others present in the environmental pool.

The *Aeromonas* communities retrieved from *I. lusitanicum* individuals varied across sampled locations and seasons, revealing specific profiles for each combination of variables. Water temperature has been proposed as a limiting factor shaping *Aeromonas* prevalence [66], and the results retrieved from the molecular typing in this study seem to corroborate this association. Similarity relationships among the isolates were highly shaped by the sampling season, as described before [66], suggesting that biotic characteristics differing among dry and wet season determine the bacterial community structure in each sampling time. Additionally, no strain permanence was observed in any location between seasons. Also, no clones were detected in different locations and the bacterial clustering was influenced by location, supporting the idea that each stream presents autochthonous populations specific to the conditions established in a determined location. In our study, however, it is not possible to determine if this fluctuation in the bacterial diversity is a sole product of temperature, since water physical and chemical parameters varied in all locations across seasons. Other factors have been implied in *Aeromonas* prevalence in aquatic streams, such as the redox potential, conductivity, organic matter level, pH, dissolved oxygen, total dissolved nitrogen, phosphate levels and water turbidity [54,55,70,74,75]. It is likely that the observed *Aeromonas* communities in *I. lusitanicum* are a result of the input of multifactorial biotic and abiotic aspects, shaping the final community structure.

A high bacterial diversity was observed in both seasons. The diversity levels differed between some locations (i.e., Laje and Jamor), revealing the influence of the presence of determined clones and hence the predominance of some strains in a given location. Although these clones were not widely prevalent across our study, they expose the success certain strains have to colonize a higher number of individuals, consequently being more prevalent. Additionally, it is possible that the conditions experienced in both locations drive the establishment of more successful bacterial clusters in relationship to adverse factors present in the streams. Colin et al. [76] exposed how urbanization of freshwater streams favours the abundance of determined bacterial groups (i.e., *Aeromonas* spp.) and leads to decreasing levels of bacterial diversity in fishes’ skin. In our study, Jamor population is the one with higher human population density in its vicinity, and the observed reduced diversity levels are a probable consequence of the stressors fish are exposed to, namely the degree of water pollution [70,76].

### 3.3. Virulence Factors

The majority of the virulence factors studied, as well as the virulence indexes, were more prevalent and higher in isolates from animals sampled in the dry season. Several factors have been proposed to modulate the regulation of bacterial virulence gene expression. Water temperature has an important role in virulence gene expression, induced by bacterial temperature sensor systems that detect temperature shifts and trigger changes in gene expression [77,78]. In mesophilic aeromonads, increases in temperature are associated with higher virulence and mortality, associated with the upregulation of specific virulence pathways and the increased production of extracellular toxins [77,79]. Nutrient availability, and specifically of nitrogenous compounds [80], influences the activation of metabolic pathways. When water presents higher organic matter, bacteria possess available energy to increase their virulence gene expression and intensify infectivity [81,82]. Contrarily, in situations of nutrient deprivation, virulence gene expression is downregulated as a measure to save energy [83]. Additionally, the exposure to stress hormones from the host (i.e., norepinephrine) increases the expression of a wide array of virulence genes [84]. It can be concluded that the environmental conditions of the dry season (i.e., warmer waters, higher nutrient load and increased fish stress due to overpopulation) result in an amplified virulence expression by *Aeromonas* spp., as demonstrated by our results.

The virulence indexes varied across the *Aeromonas* species. Globally, *A. caviae* and *Aeromonas* sp. presented the highest indexes. This finding is probably related to a sampling bias, since these groups were substantially smaller than the other sampled species. Several pan-genomic analyses performed in the genus *Aeromonas* report a hierarchization in virulence potential among the several species, related to the abundance and diversity of virulence genes [85,86,87]. Among them, *A. hydrophila* is generally considered one of the most pathogenic species, followed by *A. veronii*. *A. media*, on the other hand, is considered to display a low virulence profile [87].

Virulence in the *Aeromonas* genus results from a multifactorial array of virulence factors, including extracellular products and slime production [88]. High prevalence of lipolytic, haemolytic, DNase and proteolytic activity observed in this study are in accordance with results from previous investigations [61,68,89,90,91,92]. Similarly, low prevalence of gelatinolytic activity has also been observed before [93], as well as equivalent levels of slime production [94,95]. However, prevalence values and trends seem to vary among surveillance studies [90,95,96,97,98,99], suggesting that virulence signatures of a specific *Aeromonas* community are the result of an interaction between the community’s composition and the environmental factors shaping their habitat. This is possibly why specific associations of variations in prevalence of extracellular products and slime production and selected streams were observed in this study.

### 3.4. Antimicrobial Resistance

Although some variability can be observed, members of the genus *Aeromonas* are described as generally being resistant to penicillins, narrow spectrum cephalosporins, macrolides (clarithromycin) and antifolates (sulfamethoxazole), while presenting susceptibility to aminoglycosides, carbapenems, fluoroquinolones, extended-spectrum cephalosporins, monobactams, nitrofurans, phenicols and tetracyclines [16,23,88]. Our results corroborate these general definitions and are in accordance with previous surveys [61,63,68,69,100,101], although some variations exist for specific compounds that likely reflect local dynamics characteristic to each study area. *Aeromonas* intrinsic resistance to many β-lactam antibiotics is widely acknowledged and it is the result of the combination of the constitutive expression of a several array of β-lactamases with an external membrane with a low permeability [102]. In our study, it is noteworthy that almost a third of the isolates presented non-susceptibility to carbapenems, an antimicrobial class considered of last-resource. Tacão et al. [103] reported a high incidence of *bla*_CphA_ genes, conferring resistance to carbapenems, in *Aeromonas* spp. collected from the Vouga river basin (Portugal). The role of wild animals (i.e., nutria) in serving as reservoirs of carbapenem-resistant *Aeromonas* was previously showed [104]. Similarly, quinolone resistance is also recognized in the genus *Aeromonas* [105]. In our study, 40% of the isolates presented non-susceptibility to enrofloxacin. Our results show the establishment of antibiotic resistance phenotypes in bacteria from small riverine ecosystems in the Lisbon district. Furthermore, we stress the role of wild species, like *I. lusitanicum*, as reservoirs of clinically important resistance determinants in these environments.

In total, 30% of the isolates detected in this study were considered multidrug resistant. While the prevalence of multidrug resistant *Aeromonas* varies among studies, several authors report high levels of multidrug resistance in their surveys among this genus [24,62,69,98]. In a recent investigation with *Aeromonas* isolates collected from water samples in the river Tua (northern Portugal), Gomes et al. [106] determined that 83.3% of the studied collection was multidrug resistant. Although it is alarming that about a third of the isolates in this study present a multidrug resistance phenotype, current results suggest that sampled locations might be under lower selective pressure than other streams in the national territory and abroad. The observed variations in the resistance signatures of the isolates collected in each location and season likely reflect the demographic dynamics to which these streams are exposed and are a mirror of the resistome characteristic to the anthropogenic communities (both humans, domestic animals and crops) in close association with them.

In this study, both the MAR index scores and the prevalence of multidrug resistant strains were higher in the wet season. Some studies report an increase in antibiotic residues, antibiotic resistant bacteria (ARB) and antibiotic resistance genes (ARGs) in the dry season [107,108]. This can be linked as a direct consequence of the decrease in water levels observed in warmer months, resulting in a concentration effect in the streams. However, building evidence expose the wet season as a period when resistance determinants, especially ARGs, increase [109,110,111]. Rainfall is considered a major driver of ARGs prevalence in river streams, since rainfall events increase the transfer of resistance determinants from terrestrial settings (e.g., urban, agricultural) to water bodies through runoffs [112,113,114]. Contrary to chemical pollutants, ARGs impact is less likely to be hampered by the dilution effect caused by the rainfall [115]. In our study, the sampling performed during the wet season was preceded by the most intense raining events of the year (November–January) which possibly translated into the differences in antibiotic resistance between seasons observed. Similarly, such runoff events can also explain the differences in MAR index scores observed across isolates from different sampling locations. Higher scores were recorded in isolates from animals sampled in the Jamor and Laje rivers, two streams located in urban settings and associated with higher human population densities. Human-impacted environments play an important role in increasing and promoting the transmission of ARGs in rivers, and the prevalence of ARGs increases with urbanization [116]. Specific anthropogenic activities, such as agricultural runoffs, urban wastewater discharging and antimicrobial therapy use, promote a selective pressure in the microbiota present in that environment, shaping its antibiotic resistance signatures [115]. It seems that the resistome of a river is highly influenced by its level of urbanization, revealing the risk that urban rivers pose as reservoirs of resistance in aquatic environments.

## 4. Materials and Methods

### 4.1. Sampling Site Description and Fish Sampling

The selection of sampling sites followed a set of criteria to fulfil study inclusion. Namely: (1) the current known distribution of *I. lusitanicum*, established based on annual monitoring census performed by our team [6]; (2) the level of water abstraction in the potential sampling sites, accessed two weeks prior to sampling by on-site observations, to estimate fish presence; (3) different levels of water quality and environmental parameters between basins (pH, temperature, dissolved oxygen, total dissolved solids, electrical conductivity, nitrites, nitrates, phosphates), as determined from previous monitoring census (Sousa-Santos, personal communication); (4) different levels of mean human population density in the civil communities closely associated with the sampling sites, as determined by the Census 2011 Program [117]. In total, four river basins were selected (Lizandro, Samarra, Jamor, Laje), representing different examples of rural and urban environments with distinct water quality conditions. Sampling was performed in previously characterized sites (Lizandro: 38.886701°, −9.298140°; Samarra: 38.894761°, −9.433734°; Jamor: 38.720832°, −9.249696°; Laje; 38.709159°, −9.314079°; Figure 6) [6]. Additionally, to compare differences among bacterial prevalence and skin alteration scores in the dry and in the wet season, two sampling actions were conducted in October 2017 and February 2018, respectively.

*I. lusitanicum* was used as a model species to study *Aeromonas* prevalence in wild endangered leuciscid species. *I. lusitanicum* is a leuciscid fish, endemic to Portugal, only occurring in small coastal streams of the West Region and in the larger Tagus and Sado river basins. Its selection was based on the following criteria: (1) presenting a relevant conservation threat status (critically endangered), as defined by the IUCN [118], representing the potential of bacterial infections to disrupt imperilled leuciscid populations; and (2) the occurrence in distinct stream types throughout the species’ distribution area, aiming to evaluate the influence of ecological and anthropogenic factors on *Aeromonas* epidemiology and antimicrobial resistance and virulence profiles.

The sampling protocol followed was similar at each sampling site, involving electrofishing by two operators. Fishing period was timed (from the onset of electrofishing until the collection of the last individual). One operator surveyed the stream in parallel transects in the downstream–upstream direction and collected *I. lusitanicum* individuals following standard electrofishing procedures [119]. As the animals were collected, the second operator transferred them to a container with water collected at site to allow monitoring of the stunning recuperation (consequence of the electrofishing process) and guarantee fish survival.

After recuperation, all collected animals were individually inspected for the presence of lesions commonly associated with *Aeromonas* spp. infection (i.e., haemorrhagic lesions and skin ulcers). The prevalence of lesions was recorded for each individual. A set of sixteen individuals displaying lesions was randomly selected in every sampling location for further analysis and lesion location and description was recorded. Each individual was transferred to a measuring device to allow measuring of fork length (measured from the tip of the snout to the notch of the caudal fin) and taking pictures from both sides (when possible) to record body condition and lesions extent. In each set, two age groups with 8 individuals were created based on fork length (juveniles < 40 mm; adults ≥ 40 mm). Sex was also determined by visual inspection, which was possible for some adults during the wet season (prior to the breeding season, when sexual dimorphism is more evident: females exhibit swollen abdomens and males show nuptial tubercules at the top of their heads). Finally, water excess was wiped from each individual’s body using a sterile gauze and a bacteriological sample was collected from skin areas with typical lesions using an ESwab™ LiquidAmies Collection and Transport System (ThermoFisher Scientific^®^, Waltham, MA, USA). After this step, the full recuperation of the animal was confirmed (normal swimming behaviour and respiratory rate), following by its release into the river stream. Swabs were stored at 4 °C until further processing. Bacterial isolation was performed at the Laboratory of Microbiology and Immunology of the Faculty of Veterinary Medicine, University of Lisbon, Portugal.

Habitat conditions at each sampling site were characterized regarding connectivity (i.e., fishes isolated in summer pools vs. fishes present in stretch with hydrological connectivity) and physical and chemical parameters: (1) pH, temperature, total dissolved solids and electrical conductivity; using a portable waterproof pH meter model HI98130 (Hanna Instruments^®^, Woonsocket, RI, USA); (2) dissolved oxygen, using a waterproof oxygen meter model 9146-10, with probe HI76407/10F (Hanna Instruments^®^); and (3) nitrites, nitrates and phosphates, using colorimetric strips (ITS Thorsten Betzel^TM^, Hattersheim, Germany).

### 4.2. Body Condition Score (BCS)

Photographs of lateral views from each individual were used to access the BCS using a visual score ranging from 1 to 5 produced for adult zebrafish [120], with minor adaptations. The analysis was based on two anatomical references, as suggested by Clark et al. [120], and included: (1) the relationship between the width of the cranial region between the eye and the operculum and the abdominal girth at a halfway position between the pectoral fin and the dorsal fin; and (2) the curvature of the ventral surface of the body. While fat deposition in the ventral surface of the zebrafish occurs more cranially [120], in *I. lusitanicum* the deposition is homogeneous and the ventral surface does not create a protrusion. When available, a score was given for each lateral view and an average score from both sides was produced; otherwise, scores resulted from single lateral views. Female individuals sampled in the wet season were excluded from the analysis since BCS was likely altered by egg development.

### 4.3. Skin Lesion Score (SLS)

Photographs of each individual were used to produce an individual skin lesion score by determining the level of lesions in the skin macroscopic morphology (namely, the presence of ulcerations and areas with haemorrhage). Each photograph was analysed by computer image software (ImageJ, NIH, Bethesda, MD, USA) and the skin lesion score was produced as such: (total area of skin presenting lesions/total body area) × 100. Fins, except for the caudal fin, were excluded from the body area under analysis since it was not possible to observe them in all animals. A score was obtained from the analysis of each lateral view of every individual (when possible) and a final average score was obtained per animal.

### 4.4. Aeromonas Isolation

Swabs collected from each individual were inoculated in tubes with 8 mL of Brain Heart Infusion (BHI) broth (VWR, Radnor, PA, USA), vortexed, and incubated at 37 °C for 24 h. Samples were transferred to Glutamate Starch Red Phenol (GSP) agar plates supplemented with 100,000 IU sodium penicillin g/L (Merck, Kenilworth, NJ, USA), a selective and differential agar medium for *Aeromonas* spp., and incubated at 37 °C for 12 h. Large (2–3 mm) yellow colonies surrounded by a yellow zone were presumptively identified as *Aeromonas* spp. and four distinct colonies were randomly selected for each individual and isolated into pure cultures in BHI Agar (37 °C for 24 h). Isolates were characterized regarding Gram-staining and oxidase activity and stored in Buffered Peptone Water (VWR) with 20% glycerol at −80 °C during the study. *Aeromonas hydrophila* ATCC 7966 was used as a positive control.

### 4.5. Genomic Typing

Bacterial genomic DNA was obtained by the boiling method as described in Talon, Mulin and Thouverez [121]. Genomic typing of the isolates was conducted using a Random Amplified Polymorphic DNA (RAPD) technique. The method was used as described before [122,123], with minor modifications. Primers AP3 and AP5 [122] were used in independent mixtures to perform isolates’ typing (STABVIDA, Caparica, Portugal). Each amplification reaction was performed in a final volume of 25 µL. The mixture consisted of: 12.5 µL of Supreme NZYTaq 2× Green Master Mix (NZYTech, Lisbon, Portugal), 8.5 µL of PCR-grade water (Sigma-Aldrich, St. Louis, MO, USA), 2.5 µL of Bovine Serum Albumine (0.01%; Thermo Fisher Scientific), 0.5 µL (1 µM) of primer and 1 µL of template DNA. Thermocycler conditions were as follow: 94 °C for 5 min; 40 cycles of 94 °C for 45 s, 40 °C for 1 min, and 72 °C for 2 min; and 72°C for 5 min.

Amplification products were resolved using agarose gel electrophoresis with 1.5% (*w/v*) agarose in 1× TBE Buffer (NZYTech). Gels were resolved for 50 min at 90 V. NZYDNA Ladder VII (NZYTech) was used as a molecular weight marker. The visualization of gels was performed using a UV light transilluminator and images were recorded trough the Bio-Rad ChemiDoc XRS imaging system (Bio-Rad Laboratories, Hercules, CA, USA).

### 4.6. Aeromonas Species Identification

A multiplex PCR protocol, previously described by Persson et al. [124], discriminating between *A. caviae*, *A. media*, *A. hydrophila* and *A. veronii* based on *gyrB* and *rpoB* genes was used, with minor modifications. *A. caviae* ATCC 1976, *A. hydrophila* ATCC 7966, *A. media* ATCC 33907 and *A. veronii* ATCC 35624 were used as positive controls.

PCR mixtures were performed in a final volume of 25 µL and were composed of: 12.15 µL of Supreme NZYTaq 2× Green Master Mix (NZYTech), 10 µL of PCR-grade water (Sigma-Aldrich), 0.25 µL (0.5 µM) of primers A-cav, 0.225 µL (0.45 µM) of primers A-hyd, 0.1 µL (0.2 µM) of primers A-med, 0.075 µL (0.15 µM) of primers A-ver, 0.025 µL (0.05 µM) of primers A-16S; and 1.5 µL of template DNA. Thermocycler (VWR) parameters included hot start at 95 °C for 2 min; followed by six cycles of denaturation at 94 °C for 40 s, annealing at 68 °C for 50 s, and extension at 72 °C for 40 s; and 30 cycles at 94 °C for 40 s, 66 °C for 50 s, and 72 °C for 40 s.

PCR products were resolved by agarose gel electrophoresis as described before. Gels were resolved for 45 min at 90 V. NZYDNA Ladder VI (NZYTech) was used as a molecular weight marker. Gels were visualized and recorded as described before.

### 4.7. Virulence Factors’ Evaluation

Virulence factor expression was evaluated using phenotypical assays according to protocols previously described, with minor modifications. Specifically, haemolytic activity was accessed using Columbia agar supplemented with 5% sheep (VWR) for 24 h [125]; lipase activity was determined using Spirit Blue Agar (Difco, Franklin Lakes, NJ, USA) supplemented with 0.2% Tween 80 (VWR) and 20% olive oil (commercial) for 8 h [126]; gelatinase activity was evaluated using Oxoid™ Nutrient Gelatin (Thermo Fisher Scientific) for 24 h [127]; protease activity was detected using Skim Milk Agar (Sigma-Aldrich) for 24 h [128]; DNase activity was established using DNase Test Agar with Methyl Green (VWR) for 24 h [129]; and the production of slime was determined using Congo Red Agar for 72 h [130]. Incubation temperature was established based on river’s water temperature data collected during annual monitoring census performed in the summer seasons of 2017 to 2019 (Sousa-Santos, personal communication) and averaged (22 °C). This was performed in order to mimic fish’s body temperature since they are poikilothermic.

Each isolate’s virulence index was calculated based on the ratio between positive tests for virulence factors and the total amount of virulence factors evaluated [131].

The following strains were used as controls: *A. caviae* ATCC 15468 (haemolysin negative), *A. hydrophila* ATCC 7966 (DNase and haemolysin positive), *Enterococcus faecium* EZ40 clinical isolate canine periodontal disease (slime producer), *Escherichia coli* ATCC 25922 (DNase and gelatinase negative; non-slime producer), *Pseudomonas aeruginosa* Z25.1 clinical isolate diabetic foot infection (protease and gelatinase positive; lipase negative) and *Staphylococcus aureus* ATCC 29213 (lipase positive, protease negative). *P. aeruginosa* and *E. faecium* [132,133] belong to the bacterial collection of the Laboratory of Microbiology and Immunology, Faculty of Veterinary Medicine, University of Lisbon, Portugal.

### 4.8. Antimicrobial Susceptibility Testing

Antimicrobial susceptibility testing was determined using the disk diffusion technique [134], as established in the guidelines and following breakpoints of the Clinical and Laboratory Standards Institute [135,136,137]. *E. coli* ATCC 25922 was used as a quality control. The following antibiotics (Mastdiscs^®^, Mast Group, Liverpool, UK) were evaluated: amikacin (AK, 30 µg), amoxicilin/clavulanic acid (AUG, 20–10 µg), aztreonam (ATM, 30 µg), ceftazidime (CAZ, 30 µg), enrofloxacin (ENF, 5 µg), erythromycin (E, 15 µg), florfenicol (FFC, 30 µg), imipenem (IMI, 10 µg), nitrofurantoin (NI, 300 µg), streptomycin (S, 10 µg), tetracycline (T, 30 µg) and sulfamethoxazole/trimethropim (TS, 23.75–1.25 µg). Antimicrobial compound choice followed those commonly used to treat Gram-negative infections in Human and Veterinary Medicine, with a special focus in aquaculture.

Isolates were categorized as multidrug-resistant when presenting non-susceptibility (intermediate and resistant status) to at least one antimicrobial compound in three or more antimicrobial categories [138]. Multiple Antibiotic Resistance (MAR) index values were produced for each isolate and calculated as described by Krumperman [139]. Antimicrobial compounds to which *Aeromonas* spp. are considered intrinsically resistant (amoxicillin/clavulanic, erythromycin and streptomycin) were not included in the multidrug resistance characterization and in the MAR index calculation.

### 4.9. Data and Statistical Analysis

To analyse the reproducibility level of the molecular species identification, phenotypic virulence expression, antimicrobial susceptibility testing and genomic typing, a random sample including 10% replicates was used.

BioNumerics^®^ version 7.6 software (Applied Maths, Sint-Martens-Latem, Belgium) was used to evaluate genomic typing relationships. Fingerprints similarity was found based on a dendrogram calculated with the Pearson correlation coefficient. An optimization value of 0.5% was used. Cluster analysis was achieved through the unweighted pair group method with arithmetic average (UPGMA). The reproducibility value was determined as the average similarity value of all replicate’s pairs (87.6%) with patterns presenting higher similarity values considered to be undistinguishable. Regarding isolates considered clones, one was randomly selected and only distinct strains were considered for further analysis (species identification, virulence factor’s screening, antimicrobial susceptibility testing and statistical analysis).

The diversity of the typing profiles between locations in each season was evaluated using the Simpson index (D) [140]. Clones were excluded from the analysis and clusters of molecular types were formed based on an integrated evaluation of the fingerprinting profiles and the different *Aeromonas* species. The lowest similarity value by cluster was investigated and the same value was applied to each dendrogram to enable further comparisons.

Pearson’s correlations were calculated between: (1) fish size and the BCS, (2) size and the SLS, (3) the SLS and the BCS, at the fish level, and (4) the MAR index and the virulence index, at the isolate level.

Several response variables evaluated at each site and sampling season were modelled as a function of season and origin. A binomial logistic GLM was considered for the proportion of individuals with skin lesions. A beta response (continuous values ranging from 0 to 1) was considered for the Simpson index.

Additionally, some fish level variables were modelled as a function of season and origin. A linear model was considered for both (1) BCS and (2) SLS.

Several isolate level response variables were modelled as a function of season, origin and *Aeromonas* species. Using a binomial logistic GLM, the (1) multidrug resistance of an isolate (0—No, 1—Yes) and (2) the virulence factors (0—Negative, 1—Positive) were considered. Using a beta response (continuous values ranging from 0 to 1) the (3) MAR and (4) virulence indexes were considered. Also, several isolate level response variables were modelled as a function of season and origin. Using a binomial logistic GLM, antimicrobial susceptibility was considered. Using a multinomial log-linear model (package nnet, version 7.3-15) [141], the proportion of the different species of *Aeromonas* was considered. To accommodate the possible non-independence in isolate level responses across fish, a GLMM framework with fish as a random effect was implemented but the results indicated that there was a residual amount of variability associated with that random effect and changes to results would be negligible. Hence, reporting of the results was based on the simpler GLM analysis.

Regarding the influence of cluster on SLS and BCS, a gaussian GLM was applied. Since many of the bacterial clusters had a low number of observations, all clusters with less than four observations were pooled and generated as a new cluster serving as an intercept for the model. Additionally, and since the same individual fish could display strains in more than one bacterial cluster, only one isolate observation was randomly chosen for each individual. To incorporate the variability that is induced by the loss of information of using a single isolate at each iteration, this process was repeated a thousand times by sampling random isolates at each iteration. Histograms of the *p*-values obtained for each of these 1000 iterations were produced and it was assumed that significant results would lead to histograms where a large proportion of the corresponding *p*-values would be significant. The statistical analysis was done using R software [142].

## 5. Conclusions

Current results expose differences in fish fitness across four populations of *I. lusitanicum* from the Lisbon district (Portugal), suggesting that geography and habitat quality play a key role in the sustainable development of the species. Additionally, our findings stress the vulnerability *I. lusitanicum* individuals face seasonally, being the dry season a critical period for this species. This study also produces further knowledge regarding the epidemiology of *Aeromonas* spp. in this species, shedding light on their interaction with *I. lusitanicum* and how they shift seasonally and spatially. Finally, we confirmed the presence of clinically relevant antimicrobial resistance phenotypes in *Aeromonas* isolated from *I. lusitanicum*, confirming both their potential as resistance indicators in aquatic environments and the role of *I. lusitanicum* individuals as reservoirs of zoonotic antibiotic resistant bacteria in river streams of the Lisbon district.

Studying the general health of Iberian leuciscids and their interactions with potentially pathogenic *Aeromonas* spp. deepens our knowledge on the ecology of these species and the threats they experience. This is especially relevant for two reasons. First, it complies with international legislation, such as the Water Framework Directive in Europe, in its requirement to investigate methodology to evaluate the ecological status of aquatic environments using sentinel species, such as the ones used in this study. Second, it provides important knowledge regarding the signalization of leuciscids’ populations facing higher risk of extinction that can be used in a more rational species management program.

Several knowledge gaps still exist regarding the interaction between *Aeromonas* spp. and Iberian leuciscids. Further research should focus on the evaluation of susceptibility patterns to this bacterial genus by the different species of endangered Iberian leuciscids, to pinpoint the species most at risk. Similarly, a wider evaluation of the impact other possible health stressors (e.g., infectious, pollutants, stress) have in these species and how could their control be implemented in management programs should be conducted. Finally, microbial source tracking techniques should be used to determine the origin of antimicrobial resistance determinants reported in this study to prevent introduction in wild aquatic habitats and consequent establishment of reservoirs. 

## Figures and Tables

**Figure 1 antibiotics-10-00759-f001:**
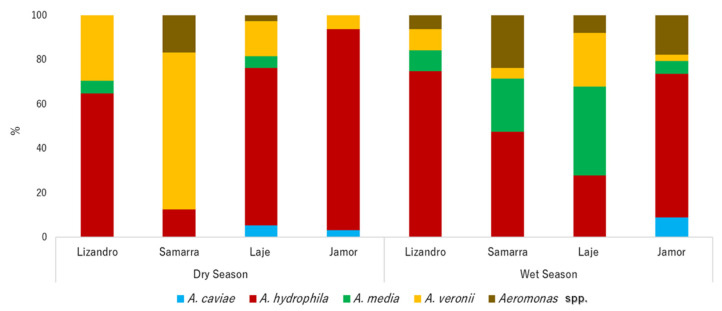
Relative abundance of *Aeromonas* species by location and sampling season.

**Figure 2 antibiotics-10-00759-f002:**
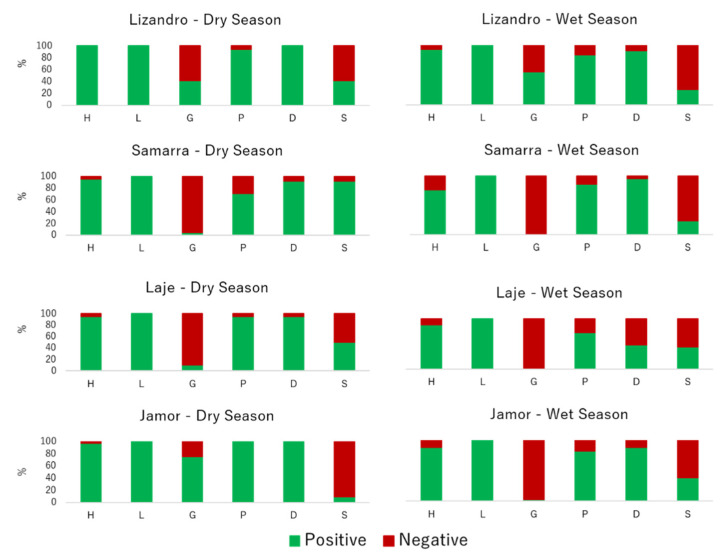
Relative prevalence of virulence factors by location and sampling season. H—hemolytic activity, L—lipase activity, G—gelatinase activity, P—protease activity, D—DNase activity, S—slime production.

**Figure 3 antibiotics-10-00759-f003:**
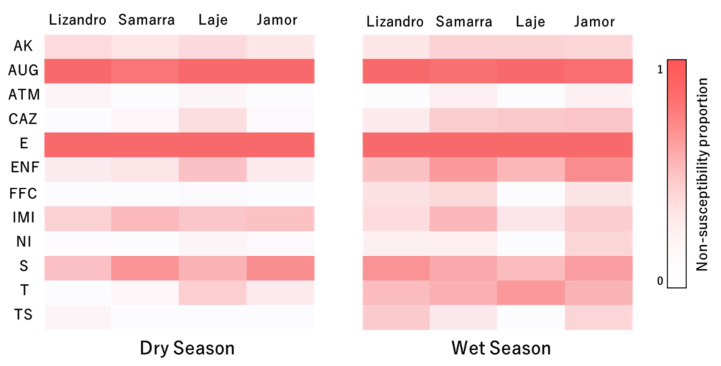
Relative prevalence of non-susceptibility to the tested antimicrobial compounds of the isolates by location and sampling season. AK—amikacin, AUG—amoxicilin/clavulanic acid, ATM—aztreonam, CAZ—ceftazidime, E—erythromycin, ENF—enrofloxacin, FFC—florfenicol, IMI—imipenem, NI—nitrofurantoin, S—streptomycin, T—tetracycline, TS—sulfamethoxazole/trimethropim.

**Figure 4 antibiotics-10-00759-f004:**
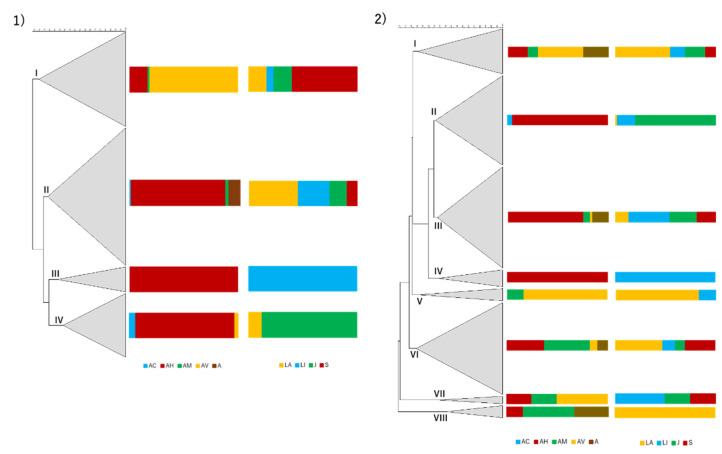
Hierarchical numerical analysis of the isolates recovered from samples collected in the dry season (**1**) and wet season (**2**). The dendrograms were based in the PCR fingerprints obtained with the AP5 and AP3 primers in a composite approach using for similarity calculation the Pearson correlation coefficient. Clustering was achieved with UPGMA. Regarding the entire dendrograms, cophenetic correlation coefficients are 0.83 for the dry season and 0.79 for the wet season. The scale used represents the percentage of similarity between the PCR fingerprints. The first column represents the dendrogram, the second the relative abundance (%) of *Aeromonas* species and the third the relative proportion (%) of isolates based on the river of origin. AC—*A. caviae*, AH—*A. hydrophila*, AM—*A. media*, AV—*A. veronii*, A—*Aeromonas* sp., LA—Laje, LI—Lizandro, J—Jamor, S—Samarra.

**Figure 5 antibiotics-10-00759-f005:**
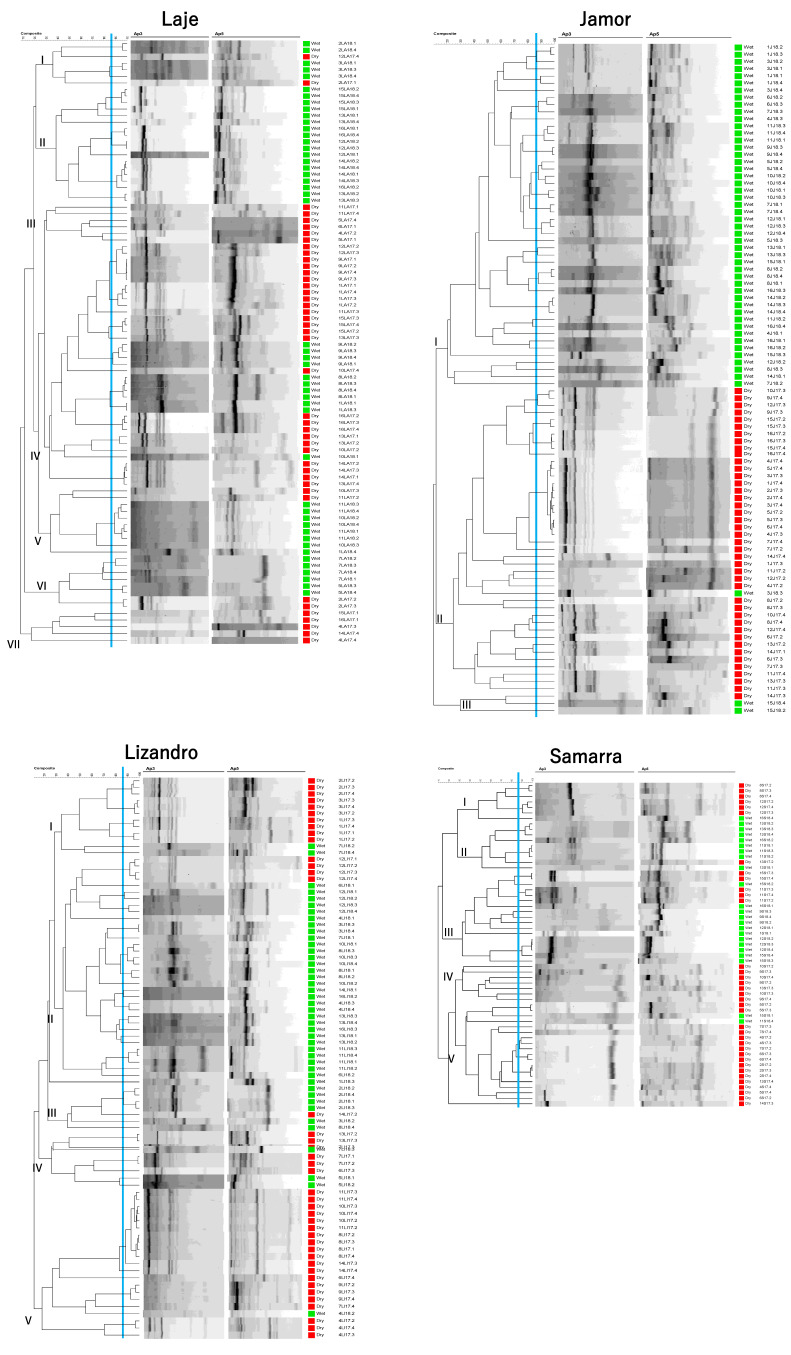
Dendrograms of isolates based on the composite analysis of RAPD fingerprints with primers Ap3 and Ap5 (Pearson correlation coefficient and UPGMA clustering) from each of the sampling locations. Cophenetic correlation coefficients are as follows: Lizandro—0.85, Samarra—0.86, Laje—0.83, Jamor—0.87. Differential coloration represents sampling season in which the isolate was collected: Red—Dry Season; Green—Wet Season.

**Figure 6 antibiotics-10-00759-f006:**
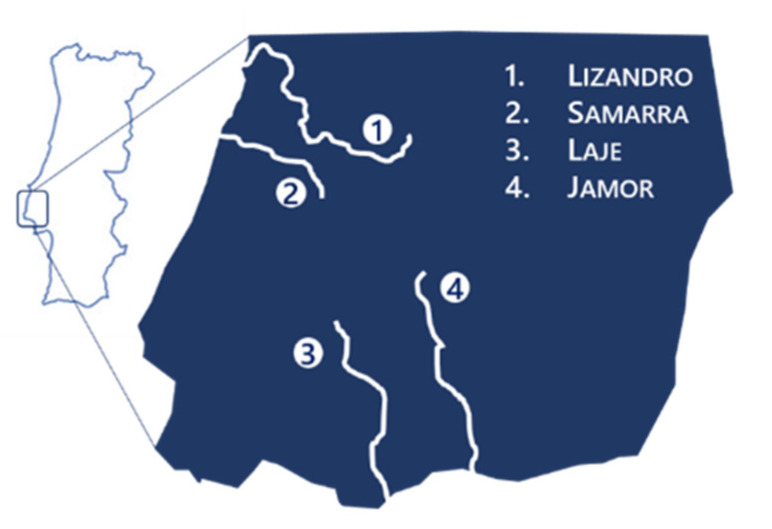
Study area.

**Table 1 antibiotics-10-00759-t001:** River’s water physical and chemical parameters and characterization regarding connectivity and human density level. C—continuous stream, I—isolated pool, R—rural, U—urban.

	Dry Season				Wet Season			
	Lizandro	Samarra	Laje	Jamor	Lizandro	Samarra	Laje	Jamor
pH	7.95	8.07	8.11	8.07	8.08	8.25	9.62	9.09
Temperature (°C)	19.4	19.9	19.9	19.2	10.9	12.9	14.2	14.9
Dissolved Oxygen (ppm)	9.58	13.41	11.74	10.39	12.9	13.69	13.63	13.59
Total Dissolved Solids (ppm)	0.41	0.57	0.53	0.33	0.44	0.49	0.43	0.3
Electrical Conductivity (mS)	0.8	1.15	1.05	0.66	0.87	0.97	0.43	0.3
Nitrites (mg/L)	0	0	0.15	0.15	0.15	0.5	0.15	0
Nitrates (mg/L)	0	0	0	0	0.5	2	0.5	0
River connectivity	C	I	I	C	C	C	C	C
Density level	R	R	U	U	R	R	U	U

## Data Availability

The data is available in this manuscript.

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
