# Peer review of "Molecular Epidemiology, Virulence Traits and Antimicrobial Resistance Signatures of Aeromonas spp. in the Critically Endangered Iberochondrostoma lusitanicum Follow Geographical and Seasonal Patterns"

_antibiotics, 2021, doi:10.3390/antibiotics10070759_

Round 1

Reviewer 1 Report

An interesting and original article, characterizing the strains of Aeromonas spp. isolated from the critically endangered wild population of Iberochondrostoma lusitanicum in a specific geographical area, is presented.

Unfortunately, I cannot recommend the manuscript for acceptance due to ambiguities in the methodology for determining susceptibility/resistance to antibacterial agents. I consider the following facts to be the most important weaknesses:

  • Why the outdated methodology according to CLSI from 2002, resp. 2013 is used?
  • Why the current EUCAST methodology was not used?
  • In the case of CLSI, testing of Aeromonas is mentioned in document M45, 3 ed. from 2015, but this document is not cited.
  • Why the primary resistance of Aeromonas to antibiotics is not respected? Susceptibility to antibiotics for which primary resistance is defined is determined, e.g. for erythromycin and amoxicillin/clavulanic acid. How the authors determined resistance/susceptibility when interpretative criteria are not specified.
  • According to which interpretive criteria was the susceptibility or resistance to streptomycin, enrofloxacin and florfenicol determined?
  • Another comment concerns clone elimination. This elimination is not described in the methodology.

For the above reasons, I recommend reworking the article and using clearly defined sources. In case of susceptibility/resistance to antibacterial agents, I recommend current criteria, including defined antibiotics, according to EUCAST.

Author Response

We thank the reviewers for their thoughtful comments and helpful critique that have contributed to improve the manuscript. Please find below a point by point reply to the reviewers where we carefully address their concerns and inputs.

Miguel Grilo

Reviewer: 1

An interesting and original article, characterizing the strains of Aeromonas spp. isolated from the critically endangered wild population of Iberochondrostoma lusitanicum in a specific geographical area, is presented.

Unfortunately, I cannot recommend the manuscript for acceptance due to ambiguities in the methodology for determining susceptibility/resistance to antibacterial agents. I consider the following facts to be the most important weaknesses:

Why the outdated methodology according to CLSI from 2002, resp. 2013 is used?

Why the current EUCAST methodology was not used?

In the case of CLSI, testing of Aeromonas is mentioned in document M45, 3 ed. from 2015, but this document is not cited.

Thank you for your questions. Used guidelines have been updated and the manuscript has been revised. The following text is now on the methods section (lines 733-735):

“Antimicrobial susceptibility testing was determined using the disk diffusion technique [134], as established in the guidelines and following breakpoints of the Clinical and Laboratory Standards Institute [135-137].”

  1. National Committee for Clinical Laboratory Standards. Performance Standards for Antimicrobial Disk and Dilution Susceptibility Tests for Bacteria Isolated from Animals; Approved Standard: 2nd edition. Document M31-A2. NCCLS, Wayne, U.S.A., 2002.
  2. Clinical and Laboratory Standards Institute. Clinical and Laboratory Standards Institute (CLSI) Performance standards for an-timicrobial susceptibility testing: 23rd informational supplement. Document M100-S23. CLSI, Wayne, U.S.A., 2013.
  3. Clinical and Laboratory Standards Institute. Clinical and Laboratory Standards Institute (CLSI) Methods for Antimicrobial Dilution and Disk Susceptibility Testing of Infrequently Isolated or Fastidious Bacteria: 3rd edition. Document M45. CLSI, Wayne, U.S.A., 2015.

Regarding guidelines choice, several aspects were taken into consideration prior to the realization of this study. Namely, and since this was a first preliminary approach regarding antimicrobial resistance surveillance in the studied environments, we aimed to characterize this habitat regarding a broader set of antimicrobial compounds that would represent several classes of antimicrobials. However, the current guidelines developed for the genus Aeromonas are restrictive in terms of choice of tested compounds, a consequence of the lack of studies and, of course, focus on the most suitable therapeutical options. We decided to follow a similar approach to other authors investigating antimicrobial resistance in Aeromonas from different settings [Figueira et al., 2011 (https://doi.org/10.1016/j.watres.2011.08.021); Igbinosa et al., 2017 (https://doi.org/10.1016/j.micpath.2017.03.037); Dahanayake et al., 2019 (https://doi.org/10.1111/jam.14355); Skwor et al., 2020 (https://doi.org/10.1128/AEM.02053-19); Gomes et al., 2021 (https://doi.org/10.3390/w13050698)], and used guidelines for other bacterial groups closely resembling Aeromonas (Enterobacteriaceae, gram negative enteric bacilli, Mannheimia haemolytica, Pasteurella multocida).

EUCAST guidelines were considered initially, but the restricted selection of testing options for Aeromonas, the possibility of using breakpoints derived from Veterinary pathogens and a more common use of CLSI guidelines in consulted literature on the topic, ultimately lead to the selection of CLSI guidelines. CLSI guidelines M100-S23 (Table 2A Enterobacteriaceae) and M31-A2 (Table 2 Veterinary Pathogens) were selected to evaluate susceptibility regarding a profile of 12 antimicrobial compounds. Although we were aware of the guidelines developed for Aeromonas (M45-A3), similarity of numerical values of these breakpoints with the ones established in M100-S23 (Table 2A Enterobacteriaceae) and less availability of options lead to the exclusion of these guidelines from our methodology. We have now included them in the methods section (line 735).

Why the primary resistance of Aeromonas to antibiotics is not respected? Susceptibility to antibiotics for which primary resistance is defined is determined, e.g. for erythromycin and amoxicillin/clavulanic acid. How the authors determined resistance/susceptibility when interpretative criteria are not specified.

Intrinsic resistances of the genus Aeromonas were ackowledged following several sources [some examples: Goñi-Urriza et al., 2000 (https://doi.org/10.1093/jac/46.2.297); Hernould et al., 2008 (https://doi.org/10.1128/AAC.01052-07); Janda & Abbot, 2010 (https://doi.org/10.1128/CMR.00039-09); Figueira et al., 2011 (https://doi.org/10.1016/j.watres.2011.08.021); Piotrowska & Popowska, 2015 (https://doi.org/10.3389/fmicb.2015.00494); Bello-López et al., 2019 (https://doi.org/10.3390/microorganisms7090363); Lim et al., 2019 (https://doi.org/10.3390/microorganisms7080224)]. Following the literature, we considered Aeromonas spp. in our study intrisically resistant to erythromycin, amoxicillin/clavulanic and streptomycin. However, and since our study had a environmental surveillance component, we found interesting to include the above mentioned antimicrobial compounds due their importance in Human and Veterinary Medicine and in order to unravell specific dynamics that could be occuring in the studied environments.

Although we studied this dynamics across origins and seasons, we did not include the results from these three antimicrobial compounds in the calculation of the MAR index values nor in the multidrug resistance characterization. In order to clarify this, we changed the method section (lines 747-750).

“Isolates were categorized as multidrug-resistant when presenting acquired non-susceptibility (intermediate and resistant status) to at least one antimicrobial compound in three or more antimicrobial categories [138]. Multiple Antibiotic Resistance (MAR) index values were produced for each isolate and calculated as described by Krumperman [139]. Antimicrobial compounds to which Aeromonas spp. are considered intrinsically resistant (amoxicillin/clavulanic, erythromycin and streptomycin) were not included in the multidrug resistance characterization and in the MAR index calculation.”

According to which interpretive criteria was the susceptibility or resistance to streptomycin, enrofloxacin and florfenicol determined?

The following guidelines were used:

  • Streptomycin - M100-S23 (Table 2A Enterobacteriaceae);
  • Enrofloxacin - M31-A2 (Table 2 Veterinary Pathogens - Gram-negative enteric bacilli);
  • Florfenicol - M31-A2 (Table 2 Veterinary Pathogens - Mannheimia haemolytica, Pasteurella multocida).

Another comment concerns clone elimination. This elimination is not described in the methodology.

The method section has been revised to accommodate this change (lines 762-764):

“BioNumerics® version 7.6 software (Applied Maths, Sint-Martens-Latem, Belgium) was used to evaluate genomic typing relationships. Fingerprints similarity was found based on a dendrogram calculated with the Pearson correlation coefficient. An optimization value of 0.5% was used. Cluster analysis was achieved through the unweighted pair group method with arithmetic average (UPGMA). The reproducibility value was determined as the average similarity value of all replicate’s pairs (87.6%) with patterns presenting higher similarity values considered to be undistinguishable. Regarding isolates considered clones, one was randomly selected and only distinct strains were considered for further analysis (species identification, virulence factor’s screening, antimicrobial susceptibility testing and statistical analysis).”

For the above reasons, I recommend reworking the article and using clearly defined sources. In case of susceptibility/resistance to antibacterial agents, I recommend current criteria, including defined antibiotics, according to EUCAST.

Reviewer 2 Report

I found the manuscript well written, comprehensive and clear.
The study includes all the epidemiological aspects of Aeromonas together with the analysis of virulence factors and antibiotic resistance of the isolated strains.
In my opinion, the abstract does not present well all that is reported in the article and I suggest implementing it.
I suggest moving or adding information about the system used (reported in L 587-589) in the introduction and abstract.
The authors write of bacterial communities associated and microbiota, but it seems to me they did not perform such studies. They searched for Aeromonas. 
Finally, how can they be sure that Aeromonas is solely responsible for skin lesions?

Author Response

We thank the reviewers for their thoughtful comments and helpful critique that have contributed to improve the manuscript. Please find below a point by point reply to the reviewers where we carefully address their concerns and inputs.

Miguel Grilo

Reviewer: 2

I found the manuscript well written, comprehensive and clear.

The study includes all the epidemiological aspects of Aeromonas together with the analysis of virulence factors and antibiotic resistance of the isolated strains.

In my opinion, the abstract does not present well all that is reported in the article and I suggest implementing it.

I suggest moving or adding information about the system used (reported in L 587-589) in the introduction and abstract.

Thank you for you comments and suggestions.

The limit for the abstract provided by this journal is 200 words (maximum) and our current version has 193 words. Although we agree with your vision, the extent of work performed in this study is difficult to fully summarize. However, we were able to include in the abstract information from lines 587-589 and keep the limit of words (line 22).

“Despite the fact that freshwater fish populations experience severe declines worldwide, our knowledge on the interaction between endangered populations and pathogenic agents is scarce. In this study, we investigated the prevalence and structure of Aeromonas communities isolated from the critically endangered Iberochondrostoma lusitanicum, a model species for endangered Iberian leuciscids, as well as health parameters in this species.”

The authors write of bacterial communities associated and microbiota, but it seems to me they did not perform such studies. They searched for Aeromonas.

We understand this critique and changed it to Aeromonas spp. when appropiate (lines 430, 810-811, 818). We still use the term microbiota or bacterial communities when discussing general concepts and not direct results from this study.

Finally, how can they be sure that Aeromonas is solely responsible for skin lesions?

Although Aeromonas spp. are commonly associated with infection in freshwater fish (both cultured and wild), the type of study performed here does not allow a definitive diagnosis. And this is because Aeromonas spp. can occur in the skin of fishes without causing infection. So, generally, definitive diagnosis of the infection rely on organ cultures (liver and kidney) and histophatological investigations. The characteristics regarding the type of species we were studying (i.e. endangered) naturally limit the type of health surveillance approach to be used.

In our study, there was a temptative step in trying to understand this connection by trying to identify relationships between the bacterial clusters and the health parameters. However, the complexity associated with our bacterial collection difficults such task and no evidences were found (as stated in the manuscript).

However, the main goal regarding this study was to evaluate the dynamics of Aeromonas associated with this endangered fish species. Because its clearly stated in the literature that the majority of these pathogens are oportunists, understading how they are shaped geografically and temporally will helps to signal possible vulnerable time-space-points that can be used to surveill in real time the stock state in the wild and to prevent future outbreaks. Similarly, it was important to understand how the virulence of the isolates varied across seasons and locations.

The evaluation of health parameters in this study is intended to allow a comparision between the state of the animal and the diversity and characteristics of the Aeromonas present in its skin.

To better clarify this assumption, we revised the manuscript (lines 316-319) to include this clarification:

“Although skin lesions in fishes can have a multifactorial origin, with several possible different agents involved, and contribution of Aeromonas spp. to the lesions observed in I. lusitanicum can’t be concluded, the methodology applied in this study can access dynamics of bacterial pathogens with relevance for Conservation Medicine and, at the same time, serve as an opportunistic basis for the study of antimicrobial resistance and virulence prevalence and temporal shifts with a particular relevance for Public Health.”

Round 2

Reviewer 1 Report

I would like to thank the authors for editing the text based on my comments, and I believe that the manuscript is now clearer and of better quality.
I recommend the article for acceptance.